# Dendritic cells loaded with FK506 kill T cells in an antigen-specific manner and prevent autoimmunity in vivo

Dana E Orange[1,2]*, Nathalie E Blachere[1,3], John Fak[1], Salina Parveen[1], Mayu O Frank[1], Margo Herre[1], Suyan Tian[4], Sebastien Monette[5], Robert B Darnell[1,3]*

[1]Laboratory of Molecular Neuro-Oncology, The Rockefeller University, New York, United States; [2]Division of Rheumatology, Hospital for Special Surgery, New York, United States; [3]Howard Hughes Medical Institute, The Rockefeller University, New York, United States; [4]Center for Clinical and Translational Science, The Rockefeller University, New York, United States; [5]Tri-Institutional Laboratory of Comparative Pathology, Memorial Sloan-Kettering Cancer Center, New York, United States

**Abstract** FK506 (Tacrolimus) is a potent inhibitor of calcineurin that blocks IL2 production and is widely used to prevent transplant rejection and treat autoimmunity. FK506 treatment of dendritic cells (FKDC) limits their capacity to stimulate T cell responses. FK506 does not prevent DC survival, maturation, or costimulatory molecule expression, suggesting that the limited capacity of FKDC to stimulate T cells may be due to inhibition of calcineurin signaling in the DC. Instead, we demonstrate that DC inhibit T cells by sequestering FK506 and continuously releasing the drug over several days. T cells encountering FKDC proliferate but fail to upregulate the survival factor bcl-xl and die, and IL2 restores both bcl-xl and survival. In mice, FKDC act in an antigen-specific manner to inhibit T-cell mediated autoimmune arthritis. This establishes that DCs can act as a cellular drug delivery system to target antigen specific T cells.

*For correspondence:
dorange@rockefeller.edu (DEO);
darnelr@rockefeller.edu (RBD)

## Introduction

FK506 (Tacrolimus) has a long record of clinical success in preventing transplant rejection and treating autoimmunity, but its use is limited by side effects including diabetes, hypertension, nephrotoxicity and neurotoxicity. FK506 is a potent inhibitor of calcineurin, a phosphatase stimulated by T cell receptor signaling to regulate the transcription factor nuclear factor of activated T cells (NFAT). Through this mechanism, FK506 inhibits IL2 production (*Bierer et al., 1990*) and triggers activated T cell death (*Horigome et al., 1998*; *Migita et al., 1999*). FK506 also limits the capacity of dendritic cells (DC) to stimulate allogeneic T cell responses in vitro (*Woltman et al., 2000*; *Szabo et al., 2001*; *Duperrier et al., 2005*). While other immunosuppressants, including rapamycin and mycophenolate, are known to inhibit DC maturation (*Hackstein and Thomson, 2004*; *Popov et al., 2006*) and thereby ameliorate autoimmunity (*Popov et al., 2006*) or promote long term transplant tolerance when transferred in vivo (*Turnquist et al., 2007*), FK506 does not inhibit DC maturation (*Woltman et al., 2000*; *Szabo et al., 2001*; *Duperrier et al., 2005*). These observations have suggested that FK506-mediated blockade of calcineurin signaling in DCs impacts T cell stimulation (*Hackstein and Thomson, 2004*). Instead, we find that FK506 treated DC (FKDC) sequester and release the drug itself, at doses sufficient to block T cell activation, to target antigen specific immune responses and to prevent collagen induced arthritis in mice. This demonstrates that DC can act as a biologic agent for drug delivery, with the potential to reduce drug dose and increase specificity in the treatment of autoimmune disease.

**eLife digest** Although our health depends on our immune system's ability to recognize and attack foreign material, this same response can cause the body to reject an organ transplant or even to spontaneously attack itself (this is called autoimmune disease). To help prevent rejection, patients who receive donated organs are given immunosuppressant drugs, with a compound called FK506, or Tacrolimus, the most commonly used. However, FK506 can have a number of serious side effects, including high blood pressure, kidney damage and diabetes.

The job of starting an immune response falls in large part to a type of white blood cell called the dendritic cell, which patrols the body in search of cells in trouble—such as those infected with viruses. Dendritic cells are efficient at engulfing dying cells, which they break down and display fragments of on their cell surface. These fragments—which are known as antigens—are presented directly to T cells, which trigger a cascade of additional immune responses leading ultimately to the destruction of infected cells.

In some cases of autoimmune disease, however, T cells begin to mistake the body's own cells for infected cells and to launch attacks against healthy tissue. Evidence suggests that immunosuppressive drugs such as FK506 can help to tone down these inappropriate immune responses. However, the use of FK506 to treat autoimmune disease has been limited by its side effects.

Now, Orange et al. have shown that dendritic cells can be exploited to deliver drugs such as FK506 in a targeted and controlled manner. When the researchers loaded dendritic cells with FK506, they found that the cells sequestered the drug and then released it slowly in quantities that were sufficient to inhibit T-cell responses for at least 72 hr.

Using a mouse model of rheumatoid arthritis—an autoimmune disease characterized by inflammation and destruction of joint tissue—Orange and co-workers demonstrated that their novel drug delivery system could be therapeutically useful. They loaded dendritic cells displaying the antigen that triggers the mouse immune system to attack joint tissue, with FK506, and used the resulting cells to treat arthritic mice. Mice that received these cells showed less severe arthritis than control animals treated with dendritic cells that had not been loaded with FK506. Moreover, the total dose of FK506 that the mice were exposed to was very low, with the result that they showed no evidence of the side effects typically seen with this drug.

This proof-of-concept study suggests that dendritic cells could be used for the gradual and controlled delivery of drugs to specific target cells within the immune system. By precisely targeting relevant immune cells, it should be possible to use much lower drug doses, and thereby reduce side effects. Follow-up studies are now required to determine whether dendritic cells can be used as vehicles for the delivery of other drugs to treat a range of diseases.

## Results

### FK506 treated DC secrete a direct T cell inhibitor

To examine the mechanism by which FK506 acts upon DCs, we treated human DC with FK506 (FKDC) for 40 hr during maturation. Despite washing to remove residual drug from the supernatants, FKDC were less effective stimulators of allogeneic T cell proliferation than DC (*Figure 1A*). Similarly, when memory T cell responses to influenza were measured by IFNγ Elispot assay (*Albert et al., 1998*; *Orange et al., 2004*), FKDC were found to be markedly less stimulatory than untreated DCs (*Figure 1B*). Moreover, in agreement with previous work (*Woltman et al., 2000*; *Duperrier et al., 2005*), DC survival, maturation, expression of MHC and costimulatory molecules were unaffected by FK506 treatment (*Figure 1C*) and therefore could not account for their reduced stimulatory capacity. We next exposed DC to FK506 for various times. Unexpectedly, FK506 pretreatment of mature DC for as little as 20 min led to the same level of T cell inhibition in the IFNγ Elispot assay as pretreatment for 40 hr (*Figure 1D*), reminiscent of previous work demonstrating maximal $^3$H-FK506 uptake by Jurkat T cells in only 20 min of FK506 treatment (*Siekierka et al., 1989b*). The allogeneic mixed lymphocyte reaction (allo-MLR) and Elispot assay require prolonged co-culture (5 days and 40 hr, respectively). To better define the molecular nature of the FKDC effect, T cells and DC were co-cultured for only 6 hr and IFNγ mRNA induction was measured by qRT-PCR. Consistent with the results from Elispot assays,

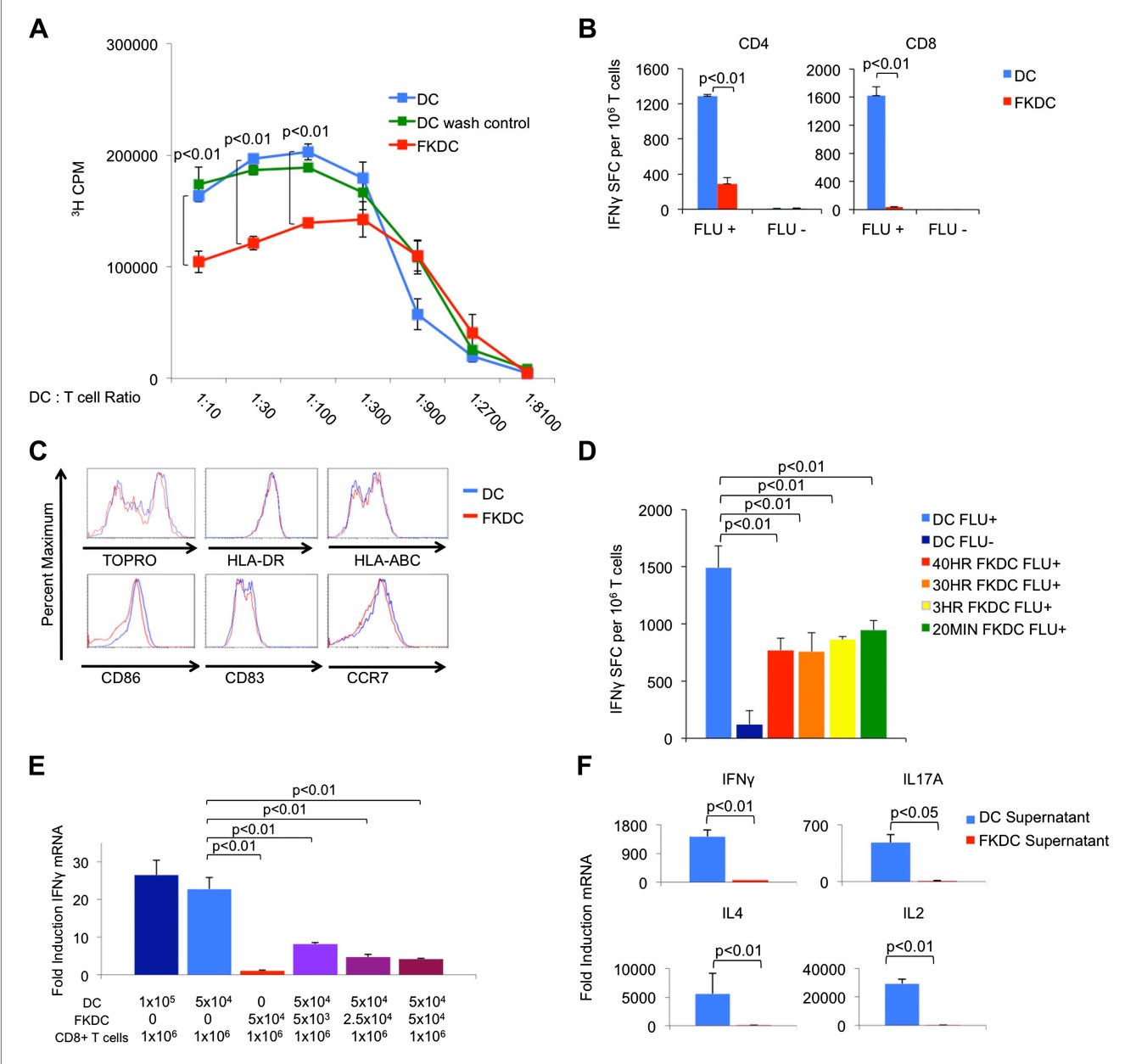

**Figure 1**. FK506 treated DC secrete a direct T cell inhibitor. (**A**) T cell proliferative responses to allogeneic DC or FKDC after 5-day coculture. DC wash control indicates untreated DCs suspended in supernatant of the last wash of FKDC. Data are mean counts per minute ± standard deviation (SD) of triplicate wells. (**B**) T cell IFNγ Elispot response to DC or FKDC presenting influenza infected (FLU+) or control (FLU−) apoptotic 3T3 cells. Data are mean spot forming cells (SFC) per million cells + SD of triplicate wells. (**C**) Phenotype of FK506 treated or untreated DC. (**D**) CD8 T cell IFNγ Elispot response to DC treated with 40 hr, 30 hr, 3 hr or 20 min of FK506. Data are mean SFC per million cells + SD of triplicate wells. (**E**) CD8 T cell induction of IFNγ mRNA in response to 6-hr stimulation with DC or FKDC. $1 \times 10^6$ T cells cultured with various ratios of syngeneic DC and FKDC (x-axis). Data are mean IFNγ mRNA induction of T cells cultured with DC presenting influenza vs DC presenting control cell + SD of triplicate wells. (**F**) Anti-CD3/28 stimulated T cells cultured in untreated DC or FKDC supernatants. Data are mean mRNA induction of stimulated vs unstimulated groups + SD of triplicate wells. p values were obtained using two-tailed unpaired t-test. Data are representative of three independent experiments.

FKDC were poor inducers of T cell IFNγ mRNA responses to influenza (**Figure 1E**). To determine if FKDC lack an activating signal or produce an inhibitory signal, untreated DC were mixed with FKDC. Combining FKDC with untreated DC inhibited the T cell stimulatory capacity of the untreated DC in a dose dependent manner (**Figure 1E**). To test whether inhibition was mediated by a secreted factor,

T cells were resuspended in DC or FKDC supernatant and stimulated with anti-CD3/28. FKDC supernatants, but not those of untreated DC, were potent inhibitors of IFNγ, IL17A, IL4 and IL2 induction (*Figure 1F*). Taken together, these data provide strong evidence that FKDC produce a T cell inhibitor.

## FK506 is the direct T cell inhibitor released by FKDC

To test if the inhibitory action of FKDC requires new gene transcription, translation or translocation through the secretory pathway, FKDC were pretreated with actinomycin, emetine, or brefeldin A prior to FK506 treatment. After washing and co-culture with T cells stimulated with anti-CD3/28, none of these treatments affected FKDC secretion of the T cell inhibitor (*Figure 2A*). We hypothesized that FK506 diffuses passively from FKDC, despite adequate washout from initial cultures. Two approaches were used to test this hypothesis: a direct measurement of FK506 from FKDC culture supernatants and a functional assay to determine if the inhibitory effect of FKDC supernatants could be abolished by removal of FK506 from the media. First, DC were treated with FK506, washed extensively, and cultured for 1–4 days, during which time they were washed daily and supernatants collected 6 hr later. FK506 was maintained in the supernatants at levels sufficient to block T cell activation for 72 hr after initial washout (*Figure 2B*). Next, to assess whether FK506 is the only inhibitor, supernatants were treated with magnetic beads coated with FK-1 antibody that binds with high affinity to FK506 and then immunoadsorbed on a magnetic column. T cells were resuspended in FK-1 depleted or control supernatants and stimulated with anti-CD3/28. Treatment of FKDC supernatants with FK-1 depleted supernatant specifically abolished the inhibitory factor completely (*Figure 2C*). To determine how FKDC might function as a repository for FK506, we evaluated DC gene expression profiles for expression of FK binding proteins. DCs express several FKBP, with FKBP12 (also known as FKBP1A) being the most robustly expressed (*Table 1*), suggesting that these proteins act in concert as an FK506 drug sink. Rapamycin is another immunomodulatory drug, which binds FKBP12, and it binds FKBP12 with higher affinity than FK506 (FK506 $K_d$ = 0.4 nM, Rapamycin $K_d$ = 0.2 nM) (*Bierer et al., 1990*). Unlike FK506, which blocks IL2 cytokine production, Rapamycin inhibits the downstream effects of IL2R signaling. The binding of FK506 and Rapamycin to a common intracellular protein was originally demonstrated in Jurkat T cells, which were rendered resistant to FK506 mediated blockade of IL2 induction when pretreated with Rapamycin (*Bierer et al., 1990*). We used this strategy to test if FKBP12 is specifically important for the accumulation of FK506 in DCs. Similar to the reported results in Jurkat cells, we found that pretreatment with Rapamycin renders FKDC unable to block IL2 induction in T cells (*Figure 2D*). This data indicates FKBP12 plays a physiologically relevant role in mediating the action of FKDC on T cell inhibition.

Taken together, these results demonstrate that DCs can absorb significant amounts of FK506, and then release drug into the supernatant in concentrations sufficient to inhibit T cell activation in a sustained manner, for at least 72 hr after a single brief treatment of DCs.

## FK506 derived from FK506 treated DC causes activated T cell death in vitro

We next evaluated the mechanism by which FKDC inhibit T cells. Previous work suggests that T cell activation when calcineurin is inhibited leads to antigen specific T cell death (*Vanier and Prud'homme, 1992*; *Migita et al., 1995, 1997*; *Horigome et al., 1998*). To examine the effect of FKDC on T cell survival, CFSE labeled T cells were cultured with allogeneic FKDC or untreated DC and apoptosis was measured by annexin V staining of undivided and postmitotic cells. There was no difference in the amount of cell death in undivided T cells (*Figure 3A*). As expected, proliferating T cells stimulated with untreated DC divided, and adding IL2 increased cell death consistent with overstimulation of the IL2 pathway and induction of activation induced cell death (AICD) (*Lenardo, 1991*). In contrast, T cells proliferating in response to FKDC were twice as likely to die and IL2 increased survival (*Figure 3B*), suggesting FK506 blocked normal T cell signaling, including IL2 production, such that cell survival was rescued by additional IL2. Since activated T cell death can be AICD-independent (*Hildeman et al., 2002*) and regulated by bcl-2 family members, we examined this pathway. Fewer proliferating T cells upregulated the survival factor bcl-xl in response to FKDC than untreated DC (*Figure 3C*), suggesting cell autonomous (mitochondrial mediated) cell death (*Akbar et al., 1996*; *Hildeman et al., 2002*), consistent with reports that FK506 sensitizes activated T cells to apoptosis by blocking bcl-xl induction

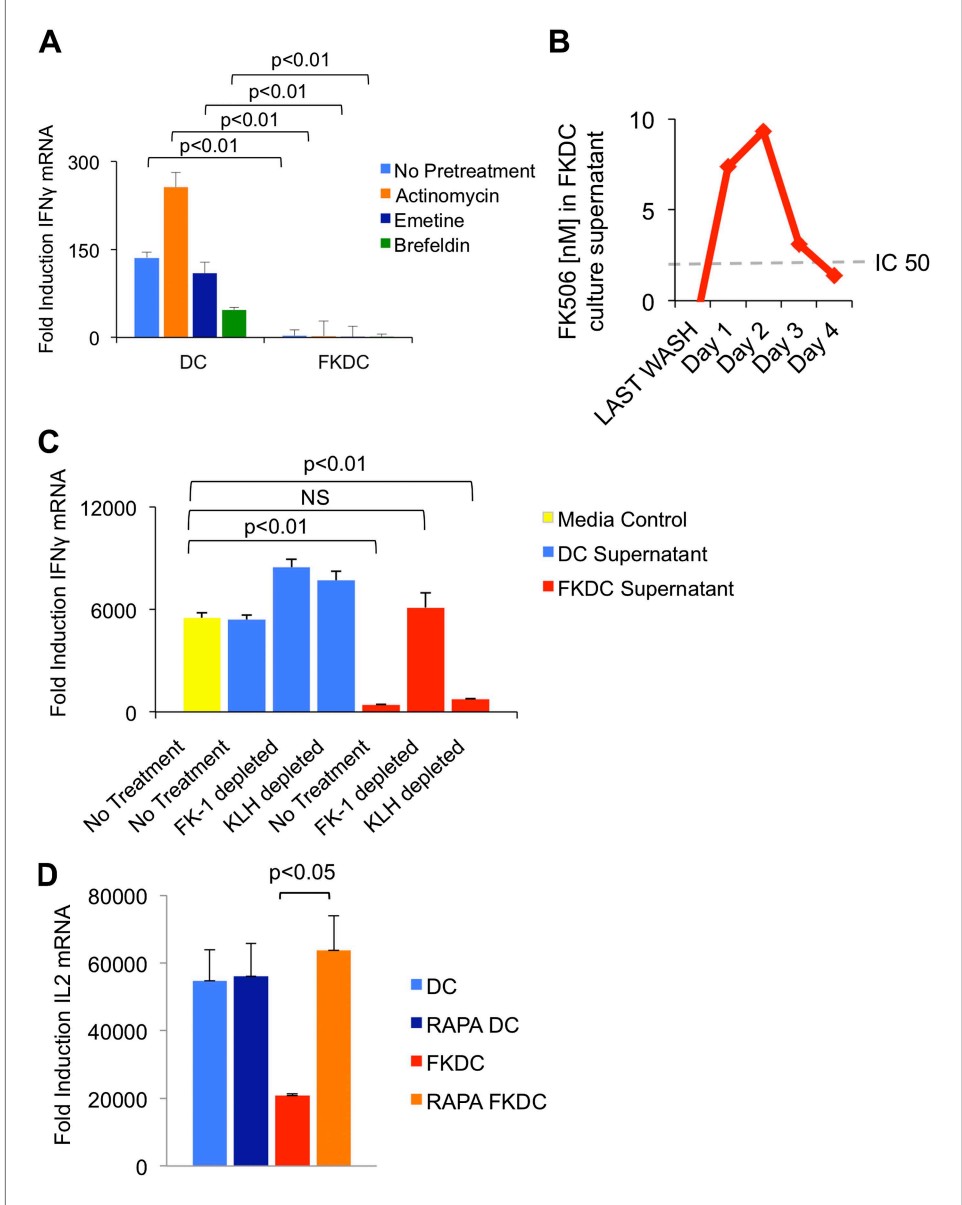

**Figure 2**. FK506 derived from FK506 treated DC blocks T cell activation. (**A**) Transcription, translation and translocation through the secretory pathway are not required for FKDC to produce a T cell inhibitor. DC were pretreated with actinomycin, emetine, brefeldin A or media prior to treatment with FK506 or media control and washed extensively. T cells were stimulated for 4 hr with anti-CD3/28 or media and mixed with treated DC. Data are mean IFNγ mRNA induction of stimulated vs unstimulated control T cell groups + standard deviation (SD) of triplicate wells. (**B**) FKDC were washed and supernatants measured daily via ELISA. Levels after last wash were undetectable. Dashed line indicates FK506 concentration that inhibits 50% induction of IL2. Data are nM ± SD of triplicate wells. (**C**) FKDC or untreated DC supernatants treated with FK-1 (anti-FK506) or anti-KLH (isotype control) antibody depletion. Depleted supernatants were added to anti-CD3/28 stimulated T cells. Data are mean IFNγ mRNA induction + SD of triplicate wells. (**D**) DCs treated with 0.5 μM rapamycin or media control for 18 hr prior to treatment with 0.5 μM FK506 or media control for 1 hr and washed extensively. Syngeneic CD4+ T cells were cultured with various treated DCs and CD3/28 beads for 4 hr. Data are mean fold induction of IL2 mRNA of stimulated vs unstimulated cells ± SD. p values were obtained using two-tailed unpaired t-test. Data are representative of three independent experiments.

**Table 1.** Mature, monocyte derived DC express mRNA of several FKBP

| Accession number | Gene name | Mean affymetrix expression level | Standard deviation |
|---|---|---|---|
| NM_000801.1 | FK506-binding protein 1A (12kD) (FKBP1A) | 1210.0 | 178.4 |
| NM_004470.1 | FK506-binding protein 2 (13kD) (FKBP2) | 370.8 | 23.7 |
| NM_003602.1 | FK506-binding protein 6 (36kD) (FKBP6) | 289.0 | 51.1 |
| NM_012181.1 | FK506-binding protein 8 (38kD) (FKBP8) | 211.8 | 82.9 |
| NM_004117.1 | FK506-binding protein 5 (FKBP5) | 203.6 | 40.8 |
| NM_002014.1 | FK506-binding protein 4 (59kD) (FKBP4) | 121.8 | 47.1 |
| NM_002013.1 | FK506-binding protein 3 (25kD) (FKBP3) | 116.5 | 21.2 |
| AF322070.1 | FK506-binding protein FKBP9 | 115.6 | 11.6 |
| NM_004116.1 | FK506-binding protein 1B (12.6 kD) (FKBP1B) | 90.7 | 25.3 |

FK506 binding proteins expressed in mature human DCs are ranked by mean Affymetrix expression level; values over 200 are considered to be robustly present. Mean expression and standard deviation are derived from four biologic replicates.

(*Migita et al., 1995*, *1997*). Moreover, we found IL2 augments bcl-xl induction and rescues T cells from death (*Figure 3C*). Furthermore, IL2 rescues IFNγ production in FKDC activated memory T cells (*Figure 3D*). In conclusion, co-culture of T cells with FKDC initiates a program of activation that leads to increased apoptosis.

## FK506 treated DC modulate antigen specific immune responses in vivo

To test the functional significance and specificity of FKDC-mediated T cell inhibition in vivo, we tested the effect of FKDC in the collagen induced arthritis animal model (CIA). DBA1/J mice were primed with intradermal bovine type II collagen (CII) emulsified in complete Freund's adjuvant (CFA), which induces a reproducible and severe arthritis. 7 and 14 days later, mice were intravenously infused with either untreated DC or FKDC loaded with either CII, or an irrelevant antigen, type I collagen (CI). Since some reports suggest FK506 may impair DC antigen presentation (*Lee et al., 2005*), DC were pulsed prior to FK506 treatment. Mice infused with FKDC pulsed with CII, but not with CI or untreated DC pulsed with CII, were significantly protected as measured by mean arthritis severity score (*Figure 4A*) and by incidence of severe arthritis (*Figure 4B*). Similar trends were observed in two other independent CIA experiments, though no formal statistical tests were performed due to small sample size (n = 3–5 per group). At day 87 after disease induction, all paws of control mice and mice treated with FKDC-CII or DC-CII were assigned histopathology (*Figure 4C*) and radiographic (*Figure 4D*) severity scores, revealing that mean arthritis severity scores were highly correlated with radiographic scores and histopathology severity scores (r = 0.97 and 0.96, respectively). Mice treated with FKDC-CII had significantly reduced radiographic (Wilcoxon rank test p<0.05) and histopathology severity scores (Wilcoxon rank test p<0.05) compared to mice treated with DC-CII.

Systemic treatment with FK506 is also effective for preventing CIA (*Takagishi et al., 1989*). In work by Takagishi et al, groups of 10–16 DBA/1 mice were immunized with intradermal bovine CII emulsified in CFA (similar to the method of arthritis induction in this manuscript). They compared 1, 2, 3 or 4 mg/kg FK506 or saline injected subcutaneously each day between day 0 and day 13. Mice treated with 4 mg/kg FK506 do not develop any arthritis; however mice treated with 1 mg/kg FK506 receive no protection from arthritis. Mice treated with 2 mg/kg on day 0–13 led to a 50% reduction in arthritis severity compared to placebo.

We compared the toxicity of our FKDC therapeutic regimen with toxicity seen using the lowest therapeutically efficacious dose (2 mg/kg) of systemic FK506. Six to 10 mice were allocated to one of three treatment groups: subcutaneous 2 mg/kg FK506 or PBS control daily for 14 days, or 0.5 million FKDC-CII intravenously on day 7 and 14. Maximal effects of either treatment regimen would be expected shortly after the last dose. Serum collected on day 14, after the last dose of either treatment regimen, was assayed for its ability to inhibit T cell activation, a standard assay for FK506 toxicity. We found that serum from mice treated with 2 mg/kg FK506, but not mice

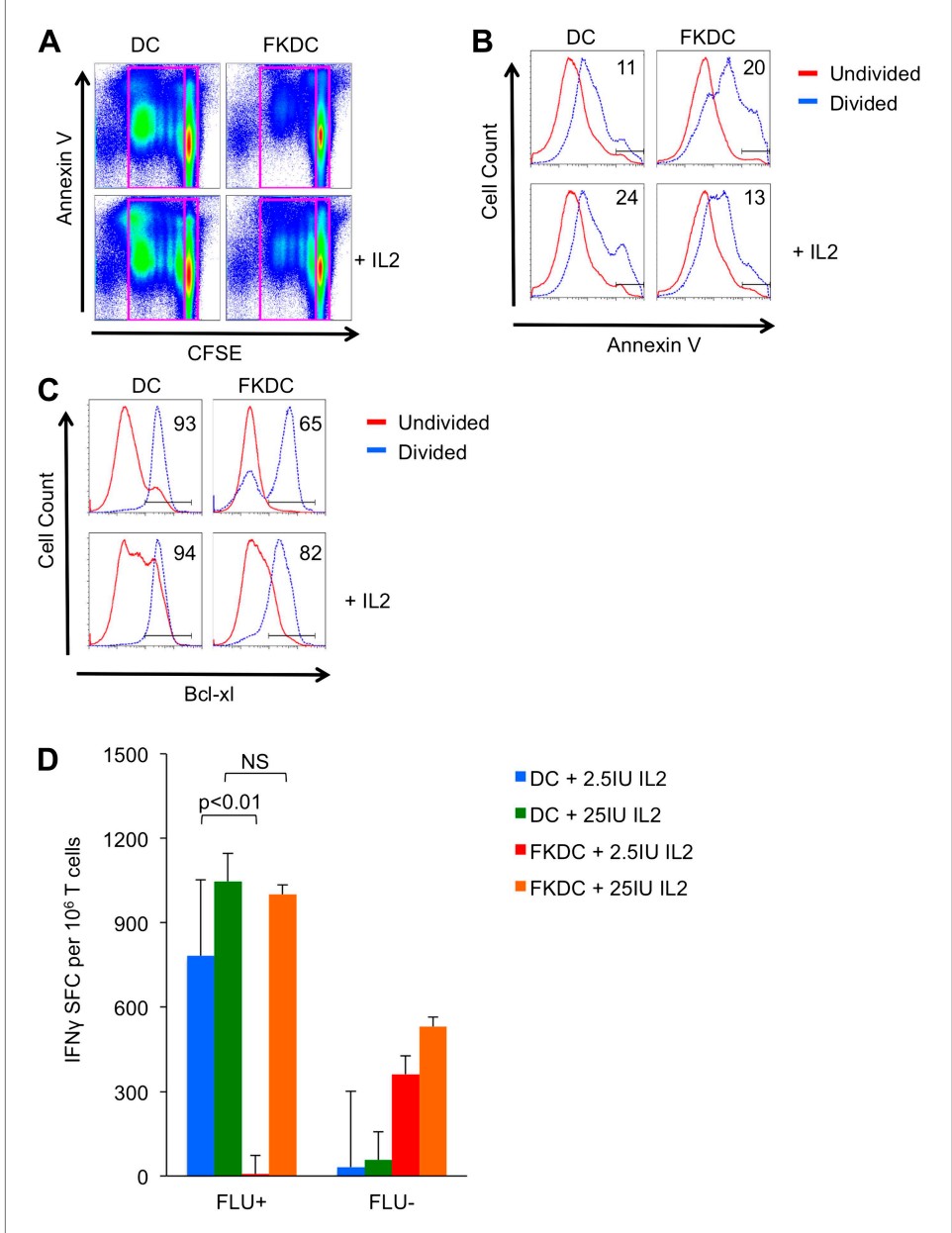

**Figure 3**. FK506 derived from FK506 treated DC causes activated T cell death in vitro. (**A**) Annexin V stained CFSE labeled allogeneic CD4+ T cells cultured with DC or FKDC ± IL2 for 4 days. Boxes indicate gating strategy for selecting undivided or divided T cells in Figure 3B and C. (**B**) Histogram of annexin V staining of undivided or divided T cells cultured with allogeneic DC or FKDC ± IL2. Percent annexin V positive divided cells. (**C**) Histogram of bcl-xl staining of undivided or divided T cells cultured with allogeneic DC or FKDC ± IL2. Percent bcl-xl positive divided cells. (**D**) CD8 T cell IFNγ Elispot response to DC or FKDC presenting FLU infected apoptotic cells supplemented with IL2. Data are mean SFC + standard deviation of triplicate wells. p values were obtained using two-tailed unpaired t-test. Data are representative of three independent experiments.

treated with FKDC, potently inhibits T cell activation (**Figure 4E**). This is consistent with higher serum levels of FK506 and systemic immunosuppression in mice treated with conventional FK506 treatment compared to DC based FK506 delivery. We also screened mice in the above treatment groups on day 14 for serum fasting glucose, amylase, lipase, BUN, AST and ALT. Amylase was increased in the systemic FK506 treatment group but not in the FKDC treatment group (**Figure 4F**). This result is consistent with prior studies revealing toxicity to the exocrine pancreas in rats

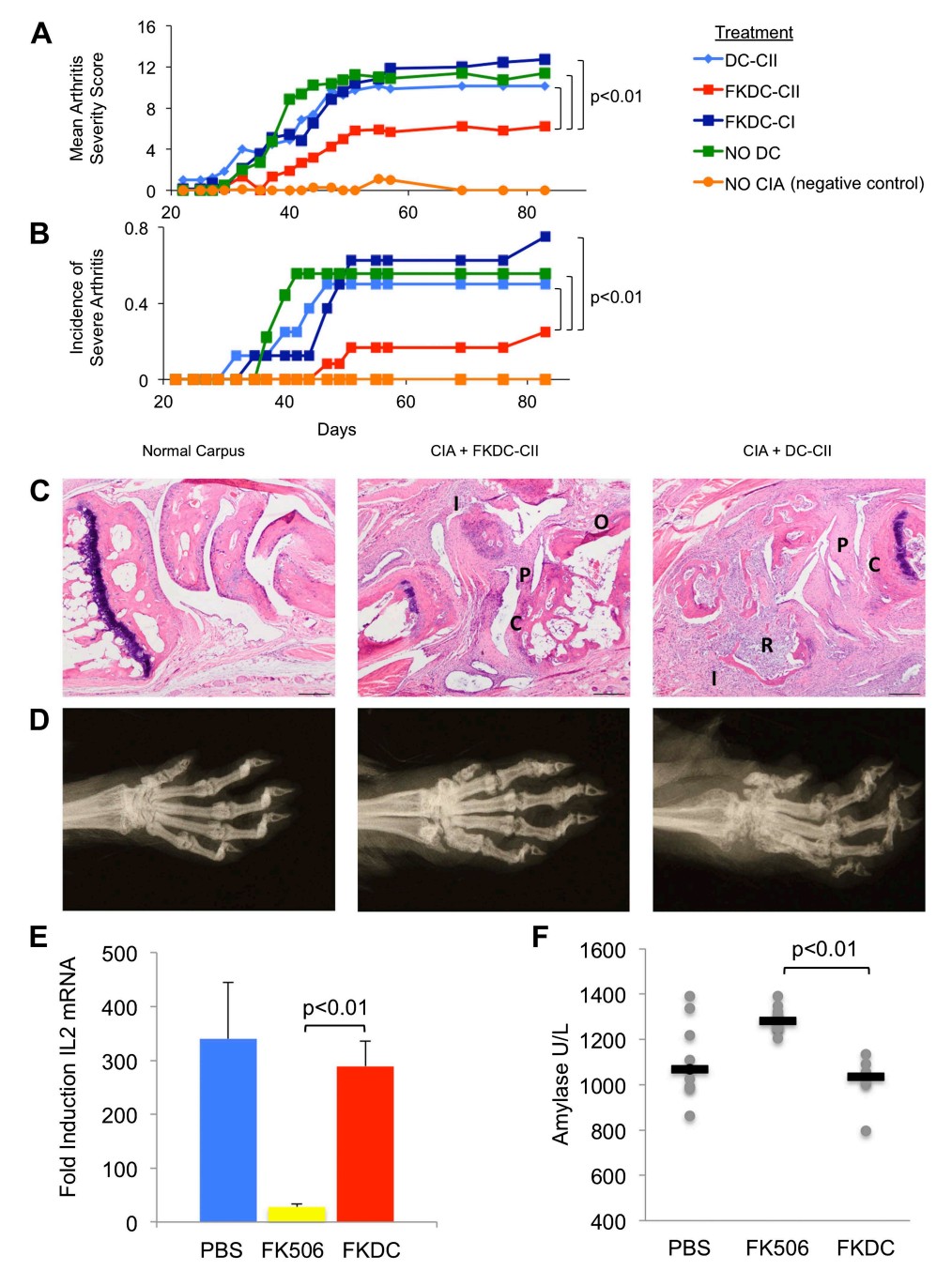

**Figure 4**. FK506 treated DC modulate antigen specific immune responses in vivo. (**A**) DBA1/J bone marrow derived DC pulsed with either type I or II collagen and treated with FK506 or media. DC were transferred intravenously 7 and 14 days after type II collagen/CFA immunization. Data are mean arthritis severity score of each group on each day. N = 8–12 per group. (**B**) Incidence of severe arthritis. (**C**) Histology of normal paw and paw of CIA mouse treated with FKDC-CII or DC-CII (mouse with median histologic score per group is presented). C: Cartilage damage; R: Bone resorption; I: inflammation; P: pannus; O: Osteophyte. (**D**) Radiographs from paws presented in (**C**). (**E**) CD4+ splenocytes from an untreated mouse were cultured with serum drawn from PBS, FK506 or FKDC treated mice after the last dose of treatment (day 14), and the CD4+ T cells then stimulated for 4 hr with CD3/28 beads. Data are mean fold induction of IL2 mRNA of stimulated vs unstimulated groups + standard deviation. N = 6–10 mice per group. p value was calculated using a two tailed t-test to compare FK506 and FKDC groups. (**F**) Serum chemistries of mice treated with 14 days, as indicated, with PBS, FK506 or FKDC, as described in (**E**). N = 6–10 mice per group. Gray circles are serum amylase values for individual

*Figure 4. Continued on next page*

*Figure 4. Continued*
mice. Black bars represent median values per group. p value was calculated using a two-tailed Mann–Whitney test comparing FK506 and FKDC treated groups. Data from (**E**) and (**F**) are representative of two independent experiments.

(*Doi et al., 1992*; *Ito et al., 1994*). There were no significant differences in the glucose, BUN, lipase, AST and ALT between any treatment groups (data not shown). We conclude that in vivo transfer of antigen pulsed FKDC modulate immune outcomes in an antigen specific manner and protect from systemic immunosuppression and off target organ toxicity.

## Discussion

We demonstrate that DCs treated with a brief pulse of FK506 sequester and then release sufficient drug to inhibit T cells for an extended period of days. The capacity of DCs to accumulate FK506 likely relates to their high expression of binding proteins such as FKBP12, which have high affinity for FK506 (*Siekierka et al., 1989a*). This suggestion is supported by the observation that T cells, which also express high levels of FKBP12, have the ability to concentrate radiolabeled FK506 500–1000 fold (*Dumont et al., 1994*). These observations also suggest the possibility that the use of such cellular drug sinks could be extended to other cell types with abundant FKBP expression, or even extrapolated to other agents with analogous intracellular binding proteins.

The actions of FKDC are potentially clinically relevant, as they prevent autoimmune arthritis in mice. The lack of efficacy of FKDC loaded with irrelevant antigen indicates that the amount of drug delivered by FKDC is not by itself sufficient to elicit a systemic effect. Upon encounter with antigen-specific FKDC, T cells initiate a program of activation but then undergo apoptosis, explaining the antigen specific effect of FKDC transferred in vivo. Systemic treatment of CIA with FK506 requires a minimum of 2 mg/kg or approximately 40 μg subcutaneously to elicit any therapeutic effect (*Takagishi et al., 1989*). In contrast, the amount of FK506 measured in culture supernatants (*Figure 2B*) predicts the dose delivered by $0.5 \times 10^6$ FKDC is a maximum of 5 ng; this means 8000-fold less drug is required to elicit a therapeutic effect when delivered by a DC. Since DC can be pulsed with antigen of interest prior to drug loading, DC based drug delivery has the potential to limit adverse events by substantially limiting required doses by precisely targeting antigen specific T cells. Antigen-loaded DC have been widely used with excellent safety profiles (*Sabado and Bhardwaj, 2010*), and this platform may be of interest to explore in clinical studies.

## Materials and methods

### FK506 treatment of DC
Human CD14+ cells cultured with GM-CSF and IL4 for 6 days were matured with TNF-α and PGE-2. DC were cultured with 0.5 μM FK506 (Astellas Pharmaceuticals, Killorglin, Ireland) or media control at 37°C for various intervals and washed.

### Allo-MLR
DC or FKDC were cultured in triplicate wells with $2 \times 10^5$ allogeneic CD4 T cells for 5 days and loaded with 4 Ci/ml H-thymidine (*Migita et al., 1999*) for 18 hr. T cells were assayed for beta emission and c.p.m. were measured.

### Elispot assay for IFNγ release
Uninfected or influenza infected 3T3 cells were exposed to ultraviolet irradiation to induce apoptosis and subsequently cultured with immature DC and maturation stimulus as above. DC were treated with FK506 or media for various durations. DC were washed extensively and co-cultured with syngeneic CD4 or CD8 T cells for 40 hr (1 DC: 30 T cells) in a IFNγ antibody coated 96-well microtiter plate.

### qRT-PCR assay for induction of cytokine mRNA
mRNA was isolated from T cells cultured with DC for 6 hr with an RNeasy kit (Qiagen, Valencia, CA), cDNA was synthesized with Quantitect reverse transcription kit (Qiagen), including genomic DNA removal. Primer pair sequences: Human IFNγ Forward: TCAGCTCTGCATCGTTTTGGGTTC, Reverse: TCCGCTACATCTGAATGACCTGCAT; Human IL17A Forward: CGGACTGTGATGGTCAACCTGA,

Reverse: GCACTTTGCCTCCCAGATCACA; IL4 Forward: CCGTAACAGACATCTTTGCTGCC, Reverse: GAGTGTCCTTCTCATGGTGGCT; IL2 Forward: AGAACTCAAACCTCTGGAGGAAG, Reverse: GCTGTCTCATCAGCATATTCACAC; HRP14 Forward: CGGAGCTGACCAGACTTTTC, Reverse: GGTTCGACCGTCATACTTCTTC. Specificity (melting-curve analysis) and priming efficiency was confirmed. Stratagene Mx3000P system and PerfeCTa SYBR Green SuperMix (Quanta Biosciences, Gathersburg, MA) were used for real-time PCR.

## FK506 depletion from DC supernatants

Magnetic dynabeads (Dynal, Life Technologies, Carlsbad, CA), coated with 100 nM FK-1 (Abcam, Cambridge, MA) or anti-KLH antibody, were incubated with supernatants for 2 hr and depleted on magnetic columns.

## Blocking DC transcription, translation, or protein translocation

DC were pretreated with actinomycin (5 µM; Sigma, Milwaukee, WI), emetine (1 µM; Sigma), or brefeldin A (10 µg/ml; Sigma) for 1 hr prior to FK506 treatment. After 2 hr of treatment with FK506 or media control, FKDC or DC were washed and added to T cells stimulated with anti-CD3/CD28 stimulator beads (Invitrogen) (25 µl per $1 \times 10^6$ cells) at 37°C for 4 hr. mRNA was isolated and qRT-PCR performed. Fold induction of cytokine mRNA was calculated as 2^-(delta CT of T cells stimulated with anti-CD3/28 and mixed with DC − delta CT of unstimulated T cells mixed with DC).

## FK506 ELISA

FKDC were washed extensively. An aliquot of the last wash was harvested and cells were cultured at $1 \times 10^6$ live cells/ml to generate supernatants 6 hr later. Cells were re-washed daily and $1 \times 10^6$ live cells/ml were replated in culture to generate supernatants 6 hr later. FK506 was measured using ELISA (USCNlife, Houston, TX). Absorbance was measured in a microtiter plate reader (Bio-Rad, Hercules, CA) (450 nm) and converted into units (ng/ml) by plotting against autoantibody titer of calibrators/standards (detection range 0.156–100 ng/ml).

## Affymetrix microarray U133A

Data were obtained from a previously reported (*Longman et al., 2007*) Affymetrix array of monocyte-derived mature DCs, and analyzed using Microarray Suite 5.0 (Affymetrix, Santa Clara, CA).

## Rapamycin competition assay

Monocyte derived DC were treated with 0.5 µM Rapamycin overnight followed by 0.5 µM FK506 or media control for 2 hr and washed extensively. Treated or untreated control DC were cocultured with CD4+ T cells and CD3/28 beads for 4 hr. PCR was performed as above.

## T cell proliferation

CD4+ T cells were stained with 2.5 µM CFSE and cultured with allogeneic DC or FKDC. On the fourth day, cells were harvested, washed, and stained with Annexin V or anti-bcl-xl antibody (Southern Biotech, Birmingham, AL).

## Treatment of CIA

8- to 10-week-old, male DBA1/J mice (The Jackson Laboratory, Bar Harbor, ME) were immunized with 100 µg bovine Type II Collagen (Chondrex, Redmond, WA) emulsified in CFA. Bone marrow derived DC (*Inaba et al., 2001*) pulsed with 50 µg/ml bovine type II collagen or bovine type I collagen (Chondrex) were subsequently treated with media or 0.5 µM FK506 for 2 hr and washed. $0.5 \times 10^6$ cells were infused intravenously 7 and 14 days post immunization. Arthritis severity score (*Brand et al., 2007*) (four paws/mouse) was rated by a blinded assessor three times weekly (n = 8–12 mice/group). On day 87 of one of three experiments, limbs were fixed in 10% buffered formalin, paraffin embedded. Radiographs (Faxitron MX-20 cabinet x-ray system) and hematoxylin and eosin stained sections were scored by a board-certified blinded veterinary pathologist as previously described (*Clark et al., 1979*; *Bendele et al., 1999*).

## Toxicity assays

DBA/1J mice were treated with either 500 µL PBS control or 2 mg/kg FK506 (Astellas) subcutaneously for 14 days, or $0.5 \times 10^6$ FKDC-CII (prepared as above) intravenously on days 7 and 14. 4 hr after the last treatment, serum from individual mice was added to CD4+ splenocytes stimulated with CD3/28 dynabeads or media control for 4 hr at 37°C. mRNA was isolated and qRT-PCR performed. Fold induction

of IL2 mRNA was calculated as 2^-(delta CT of T cells stimulated with anti-CD3/28 − delta CT of unstimulated T cells). Serum was also sent to the Memorial Sloan-Kettering Cancer Center of Comparative Medicine and Pathology for serum chemistries including BUN, amylase, lipase, glucose, AST and ALT.

## Statistics

Two-tailed, unpaired T tests were used to determine the level of significance of differences between T cells stimulated by DC or FKDC in the allo-MLR, Elispot and PCR assays. Each CIA group contained 8–12 mice. Results from pilot experiments indicated that seven mice per group were required to obtain more than 80% power to detect a 50% reduction in arthritis severity score at 0.05 level. A linear mixed model was used to model arthritis severity score of mice, then a likelihood ratio test was conducted to compare if arthritis severity score of mice vaccinated with various DC groups differed. The generalized estimating equation modeled incidence of severe arthritis in mice vaccinated with the various DC groups. Spearman's correlation coefficient was used to determine the correlation between clinical mean arthritis severity scores and radiographic scores and histopathology severity scores. The Wilcoxon rank test was used to compare radiographic and histopathology scores of the mice treated with FKDC/CII or DC/CII.

## Acknowledgements

We thank Michelle Lowes and Michael Moore for helpful editorial comments.

## Additional information

### Competing interests

Robert B Darnell. Reviewing editor, *eLife*. The other authors declare that no competing interests exist.

### Funding

| Funder | Grant reference number | Author |
| --- | --- | --- |
| Howard Hughes Medical Institute | | Robert B Darnell |
| National Institutes of Health | R01 CA093507 | Robert B Darnell |
| American College of Rheumatology REF Physician Scientist Development Award | | Dana E Orange |
| The Rockefeller University Hospital CTSA | UL1RR024143 | Dana E Orange |
| The Rockefeller University Hospital CTSA | 8 KL2 TR000151 | Dana E Orange |
| Beth and Ravenel Curry clinical scholarship | | Dana E Orange |

The funders had no role in study design, data collection and interpretation, or the decision to submit the work for publication.

### Author contributions

DEO, Conception and design, Acquisition of data, Analysis and interpretation of data, Drafting or revising the article; NEB, Acquisition of data, Analysis and interpretation of data, Drafting or revising the article; JF, Acquisition of data, Drafting or revising the article; SP, Acquisition of data, Drafting or revising the article; MOF, Acquisition of data, Analysis and interpretation of data, Drafting or revising the article; MH, Acquisition of data, Drafting or revising the article; ST, Analysis and interpretation of data, Drafting or revising the article; SM, Acquisition of data, Analysis and interpretation of data, Drafting or revising the article; RBD, Conception and design, Drafting or revising the article

### Ethics

Human subjects: Peripheral blood mononuclear cells (PBMC) were obtained from healthy volunteers at The Rockefeller University Hospital. All volunteers gave informed consent for blood donation. Approval for this study was obtained from The Rockefeller University Institutional Review Board (RDA-0269).

Animal experimentation: This study was performed in strict accordance with the recommendations in the Guide for the Care and Use of Laboratory Animals of the National Institutes of Health. All of the animals were handled according to approved institutional animal care and use committee (IACUC) protocols (11004) of The Rockefeller University.

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
