## [Author Response]

We appreciate the supportive and clear review, which prompted several additional experiments as described below. It has taken several months to bring this revised submission together, since some of the experiments described here required *in vivo* studies in animals and all experiments/data have been replicated.

** First, although it is reasonable that delivery of FK506 by DCs at much lower doses should be much safer than systemic delivery, particularly because the therapeutic effect appears to be antigen-specific, this is not actually shown. To establish this, evidence should be presented to demonstrate that at comparative levels of treatment efficacy, FKDC mediated delivery of FK506 eliminates or exhibits reduced side effects as compared to systemic delivery of FK506*.

The reviewers raise a key question that was indirectly addressed but not overtly demonstrated in the original manuscript. Our data showing the efficacy of FKDC pulsed with relevant antigen, but not irrelevant antigen, implied a therapeutic effect that cannot be explained solely by the amount of FK506 delivered by FKDC. Because the amount of FK506 delivered by FKDC is 10,000 fold less than the minimal effective dose of systemic FK506 to treat collagen induced arthritis, we hypothesized that efficacy at lower doses would translate to less toxicity. However, we did not actually demonstrate this.

One issue in considering comparative toxicity studies is that rodents are relatively resistant to the toxic effects of FK506 compared to humans. FK506 is the most commonly used drug for the prevention of transplant rejection, and hundreds of thousands of humans have been treated systemically with it. In humans, an oral dose of 0.2–0.3 mg/kg/day or a parenteral dose of 0.05 mg/kg/day is known to lead to a high incidence of adverse effects [1,2], including diabetes (up to 25%), renal insufficiency (up to 56%), hypertension (up to 62%), and liver function abnormalities (up to 36%). The toxicities seen in humans were not predicted by extensive preclinical testing in rodents. For example, comprehensive evaluation of Lewis rats demonstrated that groups receiving either 0.5, 1, 2, or 4 mg/kg/day of oral FK506 (all higher doses than are used in humans) tolerated the drug; the animals had a mild, transient weight loss and only minor histolopathologic abnormalities such as perivascular eosinophilia without vascular damage in the lung [3]. There were no reproducible changes in sections of skin, muscle, fat, intestine, liver, kidney, spleen, or heart. On the other hand, subsequent work using similar doses in beagle dogs led to remarkable emaciation, vomiting and arteritis [4,5,6,7] and death of 17% of treated baboons [9].

Although FK506 mediated organ toxicity varies across species, lymphocytes from rodents and humans are similarly affected [10] and lymphocyte mediated autoimmunity such as collagen induced arthritis responds to FK506 [8]. In work by Takagishi et al., groups of 10–16 DBA/1 mice were immunized with intradermal bovine Type II collagen emulsified in complete Freund's adjuvant (similar to the method of arthritis induction in our manuscript). They compared 1, 2, 3 or 4 mg/kg FK506 or saline injected subcutaneously each day between day 0 and day 13. Mice treated with 4 mg/kg FK506 do not develop any arthritis; however, mice treated with 1 mg/kg FK506 receive no protection from arthritis. Mice treated with 2 mg/kg on day 0–13 led to a 50% reduction in arthritis severity compared to placebo.

To address the reviewers' question, we examined the toxicity of our FKDC therapeutic regimen with toxicity seen using the lowest therapeutically efficacious dose (2 mg/kg) of systemic FK506. 6–10 mice were allocated to one of three treatment groups: subcutaneous 2 mg/kg FK506 or PBS control daily for 14 days, or 0.5 million FKDC-CII intravenously on days 7 and 14. Maximal effects of either treatment regimen would be expected shortly after the last dose. Serum collected on day 14, after the last dose of either treatment regimen, was assayed for its ability to inhibit T cell activation, a standard assay for FK506 toxicity and one in which cells from rodents and humans are similarly affected by FK506 [10]. We found that serum from mice treated with 2 mg/kg FK506, but not mice treated with FKDC, potently inhibits T cell activation (Author response image 1; included as Figure 4E in the revised article). This is consistent with higher serum levels of FK506 and systemic immunosuppression in mice treated with conventional FK506 treatment compared to DC based FK506 delivery.Author response image 1.FK506 delivered by DCs protects from systemic immunosuppression.CD4+ splenocytes from an untreated mouse were cultured with serum drawn from the indicated groups of mice after last dose of either treatment regimen, and the CD4+ T cells were then stimulated for 4 hours with CD3/28 beads. Data are mean fold induction of IL2 mRNA of stimulated vs unstimulated groups + SD. N = 6–10 mice per group. P value was calculated using two-tailed unpaired t-test. Data are representative of two independent experiments.

We also screened mice in the above treatment groups on day 14 for serum fasting glucose, amylase, lipase, BUN, AST, and ALT. Amylase was increased in the systemic FK506 treatment group but not in the FKDC treatment group (Author response image 2; included as Figure 4F in the revised submission). This result is consistent with prior studies revealing FK506 mediated toxicity to the exocrine pancreas in rats [11,12]. There were no significant differences in the glucose, BUN, lipase, AST, and ALT between any treatment groups.Author response image 2.FK506 delivered by DCs protects from off-target organ toxicity.Serum amylase drawn from the indicated groups of mice after last dose of either treatment regimen, as described in Author response image 1. N = 6–10 mice per group. Gray circles are serum amylase values for individual mice. Black bars represent median values per group. P value was calculated using 2 tailed Mann Whitney test comparing FK506 and FKDC groups.

** Second, the authors speculate that the expression of FK506 binding proteins underlies the ability of DCs to accumulate and then release therapeutic levels of FK506. This is an important general concept underlying the overall approach that should be directly tested, e.g., by knockdown of the dominant FK506 binding proteins*.

DC gene expression profiles demonstrate that DC expresses the mRNA for several FKBP (including FKBP1A/FKBP12, FKBP2, FKBP6, FKBP8, and FKBP5). This led to the hypothesis that DCs accumulate significant levels of the drug because of their abundant expression of FK506 binding proteins, the most highly expressed of which is FKBP12. Rapamycin is another immunomodulatory drug, which binds FKBP12, and it binds FKBP12 with higher affinity than FK506 (FK506 K_d_ = 0.4 nM, Rapamycin K_d_ = 0.2 nM) [13]. Unlike FK506, which blocks IL2 cytokine production, Rapamycin inhibits the downstream effects of IL2R signaling. The binding of FK506 and Rapamycin to a common intracellular protein was originally demonstrated in Jurkat T cells, which were rendered resistant to FK506 mediated blockade of IL2 induction when pretreated with Rapamycin [13]. We used this strategy to test if FKBP12 is specifically important for the accumulation of FK506 in DCs. Similar to the reported results in Jurkat cells, we found that pretreatment with Rapamycin renders FKDC unable to block IL2 in T cells (Author response image 3; included as Figure 2D in the revised submission). These data indicate that FKBP12 plays a physiologically relevant role in mediating the action of FKDC on T cell inhibition.Author response image 3.Effect of Rapamycin and FK506 on DC.Mature monocyte derived DCs were treated with 0.5 μM Rapamycin or media control for 18 hours prior to treatment with 0.5 μM FK506 or media control for one hour and washed extensively. Syngeneic CD4+ T cells were cultured with various treated DCs and CD3/28 beads for 4 hours. Data are mean fold induction of IL2 mRNA of stimulated vs unstimulated cells +/− SD. P value was calculated using two tailed, unpaired t-test to compare FKDC and RAPA FKDC. Data are representative of three independent experiments.

In summary, we believe these results enhance the assertion that DC-based drug delivery protects from systemic toxicity and is therefore a worthwhile pursuit.

**References**

1. Jain, A, et al. What have we learned about primary liver transplantation under tacrolimus immunosuppression? Long-term follow-up of the first 1000 patients. Ann Surg 230, 441-448; discussion 448-9 (1999).

2. Pirsch, JD, Miller, J, Deierhoi, MH, Vincenti, F, Filo, RS. A comparison of tacrolimus (FK506) and cyclosporine for immunosuppression after cadaveric renal transplantation. FK506 Kidney Transplant Study Group. Transplantation 63, 977-983 (1997).

3. Nalesnik, MA, et al. Toxicology of FK-506 in the Lewis rat. Transplant Proc 19, 89-92 (1987).

4. Todo, S, et al. Effect of FK506 in experimental organ transplantation. Transplant Proc 20, 215-219 (1988).

5. Todo, S, et al. Immunosuppression of canine, monkey, and baboon allografts by FK 506: with special reference to synergism with other drugs and to tolerance induction. Surgery 104, 239-249 (1988).

6. Thiru, S, Collier, DS, Calne, R. Pathological studies in canine and baboon renal allograft recipients immunosuppressed with FK-506. Transplant Proc 19, 98-99 (1987).

7. Collier, DS, Thiru, S, Calne, R. Kidney transplantation in the dog receiving FK-506. Transplant Proc 19, 62 (1987).

8. Takagishi, K, et al. Effects of FK-506 on collagen arthritis in mice. Transplant Proc 21, 1053-1055 (1989).

9. Ohara, K, et al. Toxicologic evaluation of FK 506. Transplant Proc 22, 83-86 (1990).

10. Eiras, G, et al. Species differences in sensitivity of T lymphocytes to immunosuppressive effects of FK 506. Transplantation 49, 1170-1172 (1990).

11. Ito, T, et al. Protective effects of gabexate mesilate on acute pancreatitis induced by tacrolimus (FK-506) in rats in which the pancreas was stimulated by caerulein. J Gastroenterol 29, 305-313 (1994).

12. Doi, R, Tangoku, A, Inoue, K, Chowdhury, P, Rayford, PL. Effects of FK506 on exocrine pancreas in rats. Pancreas 7, 197-204 (1992).

13. Bierer, BE, et al. Two distinct signal transmission pathways in T lymphocytes are inhibited by complexes formed between an immunophilin and either FK506 or rapamycin. Proc Natl Acad Sci U S A 87, 9231-9235 (1990).